# History of Suicide Prevention with Lithium Treatment

**DOI:** 10.3390/ph18020258

**Published:** 2025-02-14

**Authors:** Leonardo Tondo, Ross J. Baldessarini

**Affiliations:** 1Department of Psychiatry, Harvard Medical School, 116 Mill Street, Boston, MA 02478, USA; rbaldessarini@mclean.harvard.edu; 2International Consortium for Mood & Psychotic Disorder Research, McLean Hospital, Belmont, MA 02478, USA; 3Lucio Bini Mood Disorder Centers, 42 Via Crescenzio, 00,193 Rome, Italy; 4Lucio Bini Mood Disorder Centers, 28 Via Cavalcanti, 09,128 Cagliari, Italy

**Keywords:** attempt, bipolar disorders, lithium, major depressive disorder, suicide

## Abstract

Suicidal behavior is prevalent among individuals with psychiatric illnesses, especially mood, substance abuse, and psychotic disorders. Over the past several decades, lithium treatment in patients with mood disorders has been increasingly used to lower the risk of suicidal behavior. This overview considers that lithium treatment has the most abundant evidence of reducing suicidal behavior. It also examines the hypothesis that higher natural lithium levels in drinking water correlate with reduced suicide rates. We report findings from trials comparing lithium treatment with its absence, placebos, or alternative treatments for suicide prevention and address substantial challenges in such studies. The mechanisms behind lithium’s potentially protective effects against suicidal behavior remain uncertain. However, it is believed that lithium may produce anti-aggressive/anti-impulsive effects that directly contribute to anti-suicidal outcomes and mood-stabilizing effects that indirectly lead to the same results. Anti-aggressive/anti-impulsive effects may be obtained at the very low levels of lithium present in drinking water, whereas recurrence prevention may be attained at therapeutic levels.

## 1. Brief History of Lithium-Associated Suicide Prevention

This review aims to describe the various stages that led to demonstrating the efficacy of long-term treatments with lithium salts in preventing suicidal behaviors.

Goodwin and Jamison’s book *Manic-Depressive Illness* [1] reported an average risk of 19% (range: 9.0–60%) for lifetime risk of suicide-associated fatalities in patients with bipolar disorder across treatments. It is also well established that bipolar disorder and recurrent major depression have the highest standardized Mortality ratios (SMRs) for suicide out of all psychiatric disorder, averaging approximately 20 times above the international general population suicide rate of about 15/100,000/year (15/100k PEY; 0.015%/year) [2]. Such high rates have been confirmed by other studies [3,4].

Suicidal behavior is strongly associated with the depressive phase of bipolar disorder, especially when mixed (manic–depressive) features are present, so it seems plausible to suggest that antidepressants may have a beneficial role in its prevention. Newer antidepressants are less toxic and produce far fewer fatalities following overdoses than the older tricyclic and monoamine oxidase inhibitor antidepressants. As such, they are safer for people attempting suicide by acute overdose. Nevertheless, antidepressants as a class have not been associated with lower suicide risk among depressed persons, possibly because of their potential to worsen agitation or activation and induce mixed manic–depressive features [5,6]. Depressive episodes in bipolar disorders, especially with dysphoric–mixed features, are far more likely to be associated with suicidal risk than hypomania or mania (“[hypo]mania”) [7,8]. This finding is consistent with a reported protective action against suicidal behavior associated with hyperthymic affective temperament [9]. Long-term treatment with lithium has been associated with the prevention or amelioration of manic, depressive, and mixed episodes of bipolar disorder, and these clinical benefits may contribute to suicide prevention.

In 1997, we reviewed the effect of lithium treatment on suicidal behavior in major mood disorder based on findings from 28 studies (1974–1996) featuring 17,000 patients with mood disorders [10]. The rate of suicidal acts (attempts and suicides) during lithium treatment was 0.37%/year, compared to 3.20%/year without lithium treatment—an 8.65-fold difference. In 12 of these 28 reports [11,12,13,14,15,16,17,18,19,20,21,22] which compared the suicidal rates with and without lithium, the annual rates of suicidal acts were substantially lower during maintenance treatment with lithium. Despite the methodological heterogeneity of these reports, the rate of suicidal acts (±SD) among patients undergoing lithium treatment averaged 0.37 ± 0.66%/year (or per 100 patient-years) in 23 studies, compared to 3.39 ± 6.54%/year for patients without lithium treatment in 13 of the reports, corresponding to a 9.16-fold difference (paired-*t* = 2.22; *p* = 0.03).

Our review was preceded by a similar one [23], published in the now defunct journal *Lithium*, with similar findings. In the first studies hypothesizing a suicide prevention effect of lithium [24,25], in which the investigators interviewed relatives, witnesses, and medical records associated with 100 recent suicides using a questionnaire which included psychiatric and medical history. Of interest, 93/100 cases were considered mentally ill (70% with depression and 15% with alcoholism), and only 1 patient was treated with lithium. The authors proposed that, if all patients had been treated with lithium, an expected decrease in relapses would have avoided 21% of the suicides.

Among other studies in our 1997 review [10], we included a double-blind treatment trial [26] of 249 patients with bipolar disorder and 78 patients with unipolar depressive disorder, among whom 2 suicides were associated with the placebo treatment (1.28%/year) and none with the lithium treatment during 24 months of follow-up. Kay and Petterson [12] also found no suicides among 123 patients followed briefly in a specialized lithium clinic. Poole et al. [27], in a comparison of five years before versus during lithium treatment, found half as many suicide attempts during the treatment of 100 patients with various mood disorders. Lepkifker et al. [14] compared the rate of suicide in 33 patients with unipolar depressive disorder before versus during lithium maintenance treatment for an average of 8.30 years and found that 21.2% had attempted suicide before treatment (2.55%/year) compared to none during lithium treatment.

Coppen and colleagues [28] also found no suicides in 103 patients with major affective disorder attending a lithium clinic for 11 years (<0.088%/year). A later extension of this study identified 1 suicide in a total of 1519 patient-years (0.066%/year) of lithium maintenance treatment for major affective disorder [29]. Moreover, that study found a 13.8-fold higher suicide rate (0.91%/year) was found in 27 patients with untreated unipolar depressive disorder, suggesting a possible protective action of lithium in major depressive disorder [29].

Müller-Oerlinghausen and collaborators [15] designed a case–control study of suicidal risk in 68 patients with various major affective disorders and at least one suicide attempt, before, during an average of 8.00 years of lithium treatment, and following its discontinuation. They found 2.1 ± 0.2 suicide attempts before lithium treatment, 2 suicides and 4 attempts during treatment (suicidal acts: 1.10%/year), and 11 after lithium discontinuation (2.02%/year). Later, Nilsson [20] found a 4.80-times (95%CI: 1.10–12.6, *p* = 0.02) higher rate of suicides in patients who discontinued lithium compared to when they had been receiving lithium treatment.

In our study of 360 patients with type I or II bipolar disorder before, during, and following the discontinuation of long-term lithium monotherapy, the rates of suicide and life-threatening attempts were 6.40-fold lower during lithium treatment than either before lithium administration or long after lithium was discontinued [22]. Notably, however, the risk of suicidal acts increased by 20-fold within several months after discontinuing lithium maintenance treatment and later fell back to the same level encountered before lithium treatment had started. Moreover, the early suicidal risk following the discontinuation of long-term treatment with lithium was two times higher following abrupt or rapid as opposed to gradual discontinuation of lithium (over <14 vs. ≥14 days). These studies were included in a meta-analysis of 22 studies representing 5647 patients (33,473 patient-years of risk) among whom suicide was 81.8% less frequent during lithium treatment (0.159 vs. 0.875 deaths/100 patient-years), with a computed risk ratio in studies with rates off/on lithium of 8.85 (CI: 4.12–19.1, *p* < 0.0001) [3].

In a later meta-analysis of 31 clinical reports (including 5 randomized, controlled trials in which suicidal behavior was reported incidentally as an adverse event rather than as a specified outcome), we found a 4.91-fold lower risk of suicides and attempts in patients with recurrent major mood disorders treated with lithium [30].

## 2. Further Studies of Lithium Treatment and Suicidal Risk

In a meta-analysis of eight studies of patients with unipolar recurrent major depression, we found that long-term lithium treatment was again associated with a substantial reduction in the risk of suicides and attempts (by approximately 76%) among patients treated with lithium compared to alternatives, mainly anticonvulsants [31]. Three more recent reviews [32,33,34] added further support for a beneficial effect of lithium on suicidal behavior in major depressive disorder.

Studies supporting a beneficial effect of lithium in reducing suicidal risk include eight randomized controlled trials, in which suicidal behavior was again reported among the adverse events and not as an explicit, predefined outcome measure [35]. In those trials, the risk of suicide attempts and suicides was 1130/100 k PEY (100,000 person-exposure years; CI: 619–1889) among patients treated with the placebo or an alternative mood-stabilizer versus 119/100 k PEY (CI: 3.01–661) with lithium treatment, indicating a 9.50-fold difference (χ^2^ = 7.15, *p* = 0.0075). In addition, a rare randomized, placebo-controlled study with suicidal behavior as an explicit outcome measure [36] found a substantial but statistically nonsignificant difference in the rates of suicidal acts between patients with major affective disorder treated for 12 months with lithium versus a placebo, whereby all three observed suicides occurred in the placebo-treated group. Also, Koek et al. [37] found, in a study based on retrospective data from a sample of 161 military veterans diagnosed with bipolar disorder, that the rates of nonlethal suicide-related events were 9.67-times lower during long-term lithium treatment than with other mood-stabilizers given with antipsychotic drugs.

To date, reduced suicidal risk during long-term treatment with lithium in patients with bipolar disorder or recurrent major affective disorder has been found in 35 original studies included in reviews and meta-analyses [3,30,38,39,40,41,42,43,44,45]. In addition, a large study based on more than 30,000 subjects (age > 15 years) confirmed a lower rate of suicide and self-harm associated with lithium treatment [46], similarly to a Taiwanese nationwide cohort survey [47], based on nearly 26,000 participants with bipolar disorder, in which suicide accounted for 19.0% of mortality with an SMR of 26.0-times above the suicide rates in the general population. Lithium was highly significantly associated with a more than two-fold lower risk of suicide than anticonvulsants (carbamazepine, lamotrigine, or valproate) [47].

In a recent study [45], we analyzed data from a total of 13 randomized, controlled trials (RCTs) reported from 1973 to 2022. Participants had a diagnosis of either bipolar disorder or a variety of recurrent major affective disorders, excluding studies involving only unipolar major depression and those considered uninformative as lacking data on suicidal events with either treatment type. Again, suicidal behavior was not an explicit outcome measure but rather was noted among adverse events. The study involved 3836 subjects during treatment with lithium alone compared to the placebo (n = 1498) or a mood-stabilizing anticonvulsant or modern antipsychotic (n = 2338) [11,31,46,47,48,49,50,51,52,53,54,55,56].

The crude pooled rate of suicides and attempts among patients treated with lithium was 14/1498, or 0.935% [CI: 0.512–1.56], as opposed to 36/2338, or 1.54% [1.08–2.13], with alternative treatments. After correcting for a mean exposure time of 1.73 years in both groups, the rate per 100,000 person-exposure years (100k PEY) averaged 540 among patients treated with lithium versus 890 for those treated with the placebo or other treatments, indicating a 1.39-fold lower risk of suicidal behaviors after lithium administration. Data from the same 13 RCTs were also subjected to a meta-analysis (Figure 1), yielding an overall odds ratio of 0.491 [CI: 0.278–0.864], favoring lithium (*z*-score = 2.46, *p* = 0.01).

## 3. Prevention of All-Cause Early Mortality with Lithium

Suicide is the leading cause of early death in people with bipolar disorder. However, these patients also show a tendency toward early mortality due to other illnesses, particularly cardiovascular and pulmonary diseases [57,58,59,60]. In several studies conducted since the early 1990s, lithium has been found effective in reducing early mortality from all causes [34,47,61,62,63] and is probably more effectively than treatment with antipsychotics or anticonvulsants [47,64]. Lithium’s effectiveness in preventing early mortality has also been confirmed indirectly by an increase in the all-cause mortality after treatment discontinuation [65].

It is noteworthy that a reduction in early mortality due to suicide has been found despite the potentially lethal toxicity of lithium in cases of acute overdoses. Mortality from overdoses of several agents was recently ranked as follows: tricyclic antidepressants (40.7 deaths per 10,000 overdoses) > acetaminophen (25.8) > lithium (13.2) > modern antipsychotics (5.80) [66]. Lethal outcomes with acute overdoses of lithium can be avoided by vomiting or timely hemodialysis [67,68]. However, attempting suicide with lithium does not seem to be common [69], perhaps because it is not widely considered to be a lethal drug. It is also possible that suicide attempts using this agent are limited by its modulation of impulsive and aggressive behaviors [70].

## 4. Controversies About Suicide Prevention with Lithium Treatment

In the half-century of research on the potential anti-suicidal effect of lithium treatment, several ethical and methodological criticism have emerged. For instance, a randomized trial among 519 members of the US Veterans Administration (VA) was interrupted for futility after several months without suicide-related events, with rates remaining similar between the lithium and placebo treatments [56]. Rather than considering it as evidence of the failure of lithium treatment, the study should have been considered inconclusive owing to the following: [a] low serum lithium levels; [b] the use of additional uncontrolled treatments; [c] the brief duration of lithium treatment; [d] the exclusion of patients at relatively high suicidal risk; and [e] the heterogeneity of diagnoses, including unipolar depression and co-occurring substance abuse [35,71]. Of note, within one month of the end of the trial, three more suicides occurred in the placebo group and were not included in the trial data analysis. However, even with these suicides included, any effects of lithium remained nonsignificant. We commented that the trial by Katz et al. [56] did not find evidence of an anti-suicidal effect because lithium was added to other treatment regimens for groups comprising a relatively small number of mostly male veterans with complex psychopathological conditions, involving relatively brief treatment with low circulating concentrations of lithium, supporting the conclusion that their findings cannot be taken as evidence that lithium lacks anti-suicidal effects [35].

In another recent negative review of evidence on the anti-suicidal effect of lithium treatment [72], the authors found no difference in the risk of suicide between the treatment with lithium or the placebo in 12 RCTs with a total of 2578 participants. A reaction by the International Group for The Study of Lithium Treated Patients (IGSLI) [73] criticized the validity of the above review, underscoring [a] the arbitrary exclusion of trials carried out before the year 2000, [b] the selective inclusion of only some studies from an earlier meta-analysis [39], [c] the exclusion of studies on lithium versus active comparators, and [d] the inclusion of studies with no information on suicide, under the questionable assumption that there were none. Comments by Bschor et al. [73] stirred another reaction by the authors of the original review [74], claiming that the exclusion of studies before the year 2000 was justified on the basis of the unreliability of their data, whilst reporting that a sensitivity analysis also including pre-2000 data produced the same negative results. They also addressed their choice of not comparing lithium with an active comparator but only against a placebo, without providing a plausible rationale.

Evidence to test the hypothesis that lithium or other treatments may reduce suicidal risk is difficult to gather. Particularly rare are randomized controlled trials against either a placebo or an alternative active treatment with suicide-associated behavior as the defined outcome, similarly to the studies performed to support the superiority of clozapine over olanzapine in the treatment of patients with psychotic disorder and high suicidal risk, based on surrogate measures such as interventions to prevent suicide rather than suicide attempts or deaths [75,76]. Making potential fatalities an explicit outcome measure faces severe ethical and practical considerations, but surrogate outcomes such as suicidal ideation or interventions intended to prevent suicide may not be reliable substitutes for actual suicidal behavior.

Again, almost all research on treatment effects on suicidal behavior has been based on incidental reports of suicidal behavior as an adverse event rather than an explicit outcome measure. This method raises the possibility of underestimating suicidal events. In addition, such study designs are not considered adequate for the regulatory support of a therapeutic indication for a specific anti-suicidal effect.

Additional matters of concern arise in studies comparing the presence of lithium treatment to its absence, including the mismatching of other clinical interventions which may alter suicidal risk. In studies involving continued versus discontinued lithium treatment, the discontinuation itself may contribute to an elevated suicidal risk [22,77].

## 5. Lithium in Drinking Water

We found 42 research reports addressing the relationship of lithium concentrations in drinking water to local rates of suicide or violent behavior [78,79,80,81,82,83,84,85,86,87,88,89,90,91,92,93,94,95,96,97,98,99,100,101,102,103,104,105,106,107,108,109,110,111,112,113,114,115,116,117,118,119]. Of these reports, 27/42 (64.3%; indicated with an asterisk in Table 1) reported data and 30/42 (71.4%) were based on ecological studies (13 from Europe, 9 from Japan, 6 from the USA, and 1 each from Chile and Iran). Of all 42 studies, 27 of them reported data on the correlations between suicide rates or SMRs and the lithium concentration in local water. Of these, 13 (48.1%) found an inverse relationship, 8 (29.6%) a partial correlation, and 6 (22.2%) no correlation. Among the 10 reviews on this topic, including the studies already cited above, 5 (50%) reported a partial correlation, 4 (40%) an inverse correlation, and 1 (10%) no correlation (Table 1).

The significance of these findings is unclear, particularly considering the extremely low concentrations of lithium in drinking water compared to the doses and blood concentrations commonly used for treating mood disorders [67]. However, it might be that exposure to even small doses of lithium during crucial times in postnatal development or for prolonged periods may have significant effects. In addition, ingested water is not just what is drunk but also what is contained in foods [79]. Rigorous studies on this topic should include only persons always residing in the given area and consistently drinking tap water rather than bottled water. Despite inconsistencies in the studies, some investigators have been sufficiently convinced to propose the addition of lithium to drinking water, much like fluoride was added to water to reduce the risk of dental caries [107,110,115].

## 6. Conclusions

The research evidence reviewed here provides strong support for an anti-suicidal effect of long-term treatment with lithium, at least in patients with bipolar disorder and probably also in those with major depressive disorder. Moreover, studies concerning the anti-suicidal effects of lithium include evidence of its superiority to alternative treatments, including mood-altering anticonvulsants (notably, carbamazepine, lamotrigine, and sodium valproate) and both first- and second-generation antipsychotics. The relevant evidence reviewed above includes many clinical comparisons of the rates of suicides and attempts during treatment with lithium or other agents, as well as evidence from at least eight randomized, blinded, controlled trials comparing lithium with a placebo or other mood-stabilizing medicines in which suicidal behavior was noted incidentally as an adverse outcome rather than as an explicit, planned outcome measure. Studies designed with suicidal behavior as a stated outcome are rare, as they are both ethically and practically challenging, and are particularly unlikely with lithium, which has received very limited commercial support as an unpatentable mineral.

The mechanisms of lithium’s anti-suicidal effects could be associated with a direct action against aggressiveness and impulsivity, which are invariably present in suicide behaviors, or the indirect benefit of lithium in preventing mood states, especially depressive states with mixed features [8]. It has been proposed that the anti-aggressive effect can be obtained with small doses of lithium, as with highly lithiated drinking water [120], whereas the mood-stabilizing effects require therapeutic doses. The lengthy process highlighting the anti-suicidal effect of lithium in patients with bipolar disorder has certainly not been without limitations. The most significant, as previously noted, concerns the incidental nature of suicidal events in various studies in which they were considered adverse effects of the therapies. Other limitations include the different designs of the studies (randomized controlled trials or open studies), the duration of the treatments, and the lack of uniformity in the therapies compared to lithium salts. Additionally, the nomenclature associated with suicide introduces variability, as suicidal acts themselves can differ between, for instance, violent or non-violent acts, attempts with clear intent or without, and methods which are more or less lethal. Nonetheless, it is noteworthy that, despite all these potential confounding sources, even the most recent studies have consistently demonstrated the efficacy of long-term treatment with lithium salts in patients with bipolar disorder, as also shown in patients with schizophrenia. Ideally, future research on suicide prevention in patients with bipolar disorder should be conducted using a randomized controlled design, including a large number of patients followed over a sufficiently long period of time. However, these requirements face significant challenges, as suicidal behaviors are rare events and the above-proposed studies would need to span several years. For this reason, securing funding would be nearly impossible, in addition to the lack of marketing interest in lithium therapy.

## Figures and Tables

**Figure 1 pharmaceuticals-18-00258-f001:**
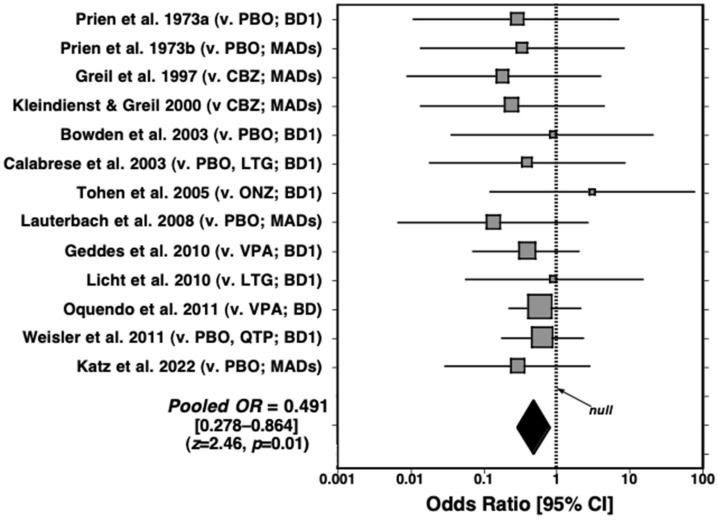
Meta-analysis of 13 randomized, controlled trials of lithium vs. placebo (PBO), carbamazepine (CBZ), lamotrigine (LTG), olanzapine (ONZ), quetiapine (QTP), or sodium valproate (VPA) given to patients with bipolar I disorder (BD1) or a mix of major affective disorders (MADs). The rates of suicide or attempts were significantly lower with lithium administration (pooled OR = 0.491; CI: 0.278–0.864; *z*-score = 2.46; and *p* < 0.01) [11,26,36,49,50,51,52,53,54,55,56]. From Tondo & Baldessarini [45].

**Table 1 pharmaceuticals-18-00258-t001:** Correlation between lithium in water and suicide or violence.

Study	Year	Country	Study Type	Results	Comment
Dawson et al. [78] *	1972	USA	Ecological	Inverse correlation	Reduced hospitalization and violence
Schrauzer & Shrestha [79] *	1990	USA	Ecological	Inverse correlation	Suicides and crimes; favors Li supplementation
Schrauzer [80]	2002		Commentary	No correlation	Lithium in food
Ohgami et al. [81] *	2009	Japan	Ecological	Inverse correlation	–––
Kabacs et al. [82] *	2011	England	Ecological	No correlation	–––
Kapusta et al. [83] *	2011	Austria	Ecological	Inverse correlation	–––
Blüml et al. [84] *	2013	USA	Ecological	Inverse correlation	–––
Giotakos et al. [85] *	2013	Greece	Ecological	Partial correlation	Lower rates of homicides, rapes, and substance abuse
Sugawara et al. [86] *	2013	Japan	Ecological	Partial correlation	Only in women
Huber et al. [87]	2014	USA	Review	–––	Varied with altitude
Giotakos et al. [88] *	2015	Greece	Ecological	Partial correlation	Homicides
Ishii et al. [89] *	2015	Japan	Ecological	Partial correlation	Only in males
Pompili et al. [90] *	2015	Italy	Ecological	No correlation	–––
Vita et al. [91]	2015	–––	Review	Partial correlation	–––
Goldstein & Mascitelli [92]	2016	–––	Review	Inverse correlation	Add Li to prevent violence
Shiotsuki et al. [93] *	2016	Japan	Ecological	Partial correlation	In men only and associated with the climate
Ando et al. [94] *	2017	Japan	Cohort	Inverse correlation	In adolescents, less depression and aggressiveness
Awad et al. [95]	2017	–––	Commentary	–––	Against adding lithium to water
Kanehisa et al. [96] *	2017	Japan	Case–control	Inverse correlation	Fewer suicide attempts
Knudsen et al. [97] *	2017	Denmark	Cohort study	No correlation	–––
König et al. [98] *	2017	Chile	Ecological	Inverse correlation	Only in some regions
Brown et al. [99]	2018	–––	Review	Partial correlation	May protect from lead exposure
Fajardo et al. [100] *	2018	USA	Ecological	Inverse correlation	Decreased all-cause mortality and loss of years
Kurosawa et al. [101] *	2018	Japan	Case–control	Inverse correlation	Lithium added to omega-3 fatty acids
Oliveira et al. [102] *	2019	Portugal	Ecological	No correlation	–––
Palmer et al. [103] *	2019	USA	Ecological	Inverse correlation	–––
Barjasteh-Askari et al. [104]	2020	–––	Review	Partial correlation	Dose-dependent effect
Del Matto et al. [105]	2020	–––	Review	Partial correlation	More in long-term lithium treatment
Eyre-Watt et al. [106]	2020	–––	Review	Inverse correlation	–––
Kozaka et al. [107] *	2020	Japan	Ecological	No correlation	–––
Memon et al. [108]	2020	–––	Review	Inverse correlation	–––
Kugimiya et al. [109] *	2021	Japan	Ecological	Partial correlation	Only in men
Liaugaudaite et al. [110] *	2021	Lithuania	Ecological	Inverse correlation	–––
Araya et al. [111]	2022	–––	Commentary	–––	Proposed a consensus
Izsak et al. [112] *	2022	Hungary	Ecological	Partial correlation	Only in women
Kawada [113]	2022	–––	Commentary	–––	–––
Liaugaudaite et al. [114] *	2022	Lithuania	Ecological	Partial correlation	Only with higher lithium levels and high rates of affective disorders
Fadaei [115]	2023	–––	Review	Inverse correlation	–––
Rihmer & Dome [115]	2023	Hungary	–––	–––	In favor of the fortification of drinking water
Bjørklund & Storchylo [116]	2024	–––	Review	Partial correlation	Critical comments on lithium supplementation
Harandi et al. [117] *	2024	Iran	Ecological	Inverse correlation	Reduced suicide attempts
Pichler et al. [118] *	2024	Switzerland	Ecological	No correlation	–––

Of the 27/42 studies reporting data (*), 13 (48.1%) found an inverse relationship between suicide rates or SMRs and lithium concentration in local water, 8 (29.6%) found partial correlations, and 6 (22.2%) found no correlation. Among the 10 reviews, including other listed studies, 5 (50%) reported a partial correlation, 4 (40%) an inverse correlation, and 1 (10%) no correlation.

## Data Availability

Not applicable.

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
