# Peer review of "History of Suicide Prevention with Lithium Treatment"

_pharmaceuticals, 2025, doi:10.3390/ph18020258_

Round 1
Reviewer 1 Report
Comments and Suggestions for Authors
This is a well-written review about the anti-suicidal effects of lithium. Nonetheless, I ask the authors to revise their manuscript following my comments to further clarify lithium effects on suicide.
The authors said “Mechanisms behind lithium’s potential protective effects against suicidal behavior remain uncertain but may involve reduced recurrence of depressive episodes, particularly those with mixed features or agitation, as well as potential benefits against impulsivity, agitation, and dysphoric mood.”. Probably, lithium may produce anti-aggressive/anti-impulsive effects, which directly contributes to anti-suicidal outcomes, while lithium may bring about mood-stabilizing effects, indirectly contributing to anti-suicidal outcomes. Moreover, the former effects can be obtained at the very low levels of lithium in drinking water, whereas the latter effects can be acquired at therapeutic levels of lithium. (Terao et al, Pharmaceuticals, 2024).
In addition, lithium may inhibit testosterone, which may decrease aggression and impulsivity, leading to suicide prevention particularly in men. (Sher, J Clin Psychiatry, 2015).
Considering the above comments, the authors should further polish their discussion.
Author Response
Comment:
This is a well-written review about the anti-suicidal effects of lithium. Nonetheless, I ask the authors to revise their manuscript following my comments to further clarify lithium effects on suicide.
The authors said “Mechanisms behind lithium’s potential protective effects against suicidal behavior remain uncertain but may involve reduced recurrence of depressive episodes, particularly those with mixed features or agitation, as well as potential benefits against impulsivity, agitation, and dysphoric mood.”. Probably, lithium may produce anti-aggressive/anti-impulsive effects, which directly contributes to anti-suicidal outcomes, while lithium may bring about mood-stabilizing effects, indirectly contributing to anti-suicidal outcomes. Moreover, the former effects can be obtained at the very low levels of lithium in drinking water, whereas the latter effects can be acquired at therapeutic levels of lithium. (Terao et al, Pharmaceuticals, 2024).
Reply in the abstract:
Mechanisms behind lithium’s potential protective effects against suicidal behavior remain uncertain but lithium may produce anti-aggressive/anti-impulsive effects directly contributing to anti-suicidal outcomes whereas its mood-stabilizing effects may indirectly contribute to anti-suicidal outcomes. Anti-aggressive/anti-impulsive effects may be obtained at the very low levels of lithium in drinking water, whereas the recurrence prevention can be obtained at therapeutic levels of lithium.
Reply in Conclusions:
Mechanisms of lithium’s antisuicidal effects could be associated with a direct action against aggressiveness and impulsivity, which are invariably present in suicide behavior or with the indirect benefit of lithium in preventing mood states, especially depressive with mixed features. It has been proposed that the anti-aggressive effect can be also obtained with small doses of lithium as it happens with highly lithiated drinking water (Terao et al. 2024), whereas for the mood-stabilizing effects, therapeutic doses of lithium are required.
Comment:
In addition, lithium may inhibit testosterone, which may decrease aggression and impulsivity, leading to suicide prevention particularly in men. (Sher, J Clin Psychiatry, 2015).
Reply:
The findings about lithium's inhibition of testosterone are too inconsistent to be included. Even less in consideration of the weak association with suicide behavior. The article cited is a case report which does not seem to pertain to our paper.

Reviewer 2 Report
Comments and Suggestions for Authors
Dear Authors
This manuscript offers a comprehensive review of lithium's role in suicide prevention among patients with mood disorders. The authors have successfully compiled substantial evidence from historical studies, clinical trials, and meta-analyses, effectively highlighting lithium's efficacy compared to alternative treatments. The review is well-written and adequately conveys the importance of lithium in suicide prevention. I have only minor revisions to suggest for improvement.
1) The introduction could better frame the objectives of the paper. I would suggest that the authors start with a clear statement of the research question or the purpose of the review.
2) The authors acknowledge the methodological variability among studies but do not sufficiently explore its impact on the reliability of the conclusions. It would be appropriate to include a detailed discussion of limitations, such as sample heterogeneity, duration of lithium treatment, and outcome definitions (e.g., suicide attempts vs. completed suicides), where possible
3) The manuscript appears, as clearly indicated in the title, overly supportive of lithium’s efficacy. However, I would suggest including a brief discussion of alternative viewpoints or evidence challenging its benefits (e.g., Nabi et al., 2022) to provide a more balanced perspective.
Author Response
Comment: The introduction could better frame the objectives of the paper. I would suggest that the authors start with a clear statement of the research question or the purpose of the review.
Reply: This review aims to describe the various stages that led to demonstrating the efficacy of long-term treatments with lithium salts in preventing suicidal behaviors.
Comment: The authors acknowledge the methodological variability among studies but do not sufficiently explore its impact on the reliability of the conclusions. It would be appropriate to include a detailed discussion of limitations, such as sample heterogeneity, duration of lithium treatment, and outcome definitions (e.g., suicide attempts vs. completed suicides), where possible.
Reply: The lengthy process that highlighted the anti-suicidal effect of lithium in patients with bipolar disorder has not been certainly without limitations. The most significant, as previously noted, concerns the incidental nature of suicidal events in various studies in which they were considered adverse effects of the therapies. Other limitations include the different designs of the studies (randomized-controlled trials or open studies), the duration of treatments, and the lack of uniformity in the therapies compared to lithium salts. Additionally, the nomenclature associated with suicide introduces variability, as suicidal acts themselves can differ—for instance, violent or non-violent acts, attempts with clear intent or without, or methods that are more or less lethal. Nonetheless, it is noteworthy that, despite all these potential sources of confounding, even the most recent studies consistently have demonstrated the efficacy of long-term treatment with lithium salts in patients with bipolar disorder, as has also been shown in patients with schizophrenia.
Comment: The manuscript appears, as clearly indicated in the title, overly supportive of lithium’s efficacy. However, I would suggest including a brief discussion of alternative viewpoints or evidence challenging its benefits (e.g., Nabi et al., 2022) to provide a more balanced perspective.
Reply: We devoted the section "Controversies about suicide prevention with lithium treatment" for alternative viewpoints also citing the study by Nabi et al.
Reviewer 3 Report
Comments and Suggestions for Authors
This manusciprt is a valuable contribution to the literature on suicide prevention and lithium treatment, presenting a strong evidence base for its effectiveness. The conclusion effectively highlights lithium’s potential as a public health intervention for suicide prevention in mood disorders. The paper is well-referenced and integrates a range of studies. Here are comments.
1.Add more specificity regarding the scope of the review, such as the types of studies analyzed (RCTs, cohort studies) and their geographical diversity.
2.Expand the discussion of why suicidal behavior was often not a predefined outcome in trials and how this impacts interpretation.
3.Include a more explicit call to action for future research, such as the need for large-scale, long-term RCTs with suicidal behavior as a primary outcome.
Author Response
Comment: Add more specificity regarding the scope of the review, such as the types of studies analyzed (RCTs, cohort studies) and their geographical diversity.
Reply: more specific aims of this review has been added in the introduction as requested by Reviewer #1. The types of studies analyzed are reported in our cited and commented meta-analysis (lines 115–118) and in another recent study (Tondo & Baldessarini 2024). To the best of our knowledge, the topic of geographical origin of the paper has not been addressed in other papers and therefore has not been included in our review.
Comment: Expand the discussion of why suicidal behavior was often not a predefined outcome in trials and how this impacts interpretation.
Reply: The topic has been addressed already (lines 231-240).
Comment: Include a more explicit call to action for future research, such as the need for large-scale, long-term RCTs with suicidal behavior as a primary outcome.
Reply: The following sentence as been added at the end of the review (lines 319-325): Ideally, future research on suicide prevention in patients with bipolar disorder should be conducted using a randomized controlled design, and the study should include a large number of patients followed over a sufficiently long period. However, these requirements face significant challenges, as suicidal behaviors are rare events, and the study would need to span several years. For this reason, securing funding would be nearly impossible, compounded by the lack of marketing interest in lithium salt therapy.